# High-Pressure-Based Strategies for the Inactivation of *Bacillus subtilis* Endospores in Honey

**DOI:** 10.3390/molecules27185918

**Published:** 2022-09-12

**Authors:** Hana Scepankova, Carlos A. Pinto, Letícia M. Estevinho, Jorge A. Saraiva

**Affiliations:** 1LAQV-REQUIMTE, Chemistry Department, Campus Universitário de Santiago, University of Aveiro, 3810-193 Aveiro, Portugal; 2Centro de Investigação de Montanha, Instituto Politécnico de Bragança, 5300-252 Bragança, Portugal; 3Laboratório para a Sustentabilidade e Tecnologia em Regiões de Montanha, Instituto Politécnico de Bragança, Campus de Santa Apolónia, 5300-253 Bragança, Portugal

**Keywords:** honey, diluted honey, preservation, high-pressure processing, hyperbaric storage, *Bacillus subtilis*, endospores, inoculation

## Abstract

Honey is a value-added product rich in several types of phenolic compounds, enzymes, and sugars recently explored in biomedical and food applications. Nevertheless, even though it has a low water activity (a_W_ ≈ 0.65) that hinders the development of pathogenic and spoilage microorganisms, it is still prone to contamination by pathogenic microorganisms (vegetative and spores) and may constitute harm to special groups, particularly by immunosuppressed people and pregnant women. Thus, an efficient processing methodology needs to be followed to ensure microbial safety while avoiding 5-hydroxymethylfurfural (HMF) formation and browning reactions, with a consequent loss of biological value. In this paper, both thermal (pressure-assisted thermal processing, PATP) and nonthermal high-pressure processing (HPP), and another pressure-based methodology (hyperbaric storage, HS) were used to ascertain their potential to inactivate *Bacillus subtilis* endospores in honey and to study the influence of a_W_ on the inactivation on this endospore. The results showed that PATP at 600 MPa/15 min/75 °C of diluted honey (52.9 °Brix) with increased a_W_ (0.85 compared to ≈0.55, the usual honey a_W_) allowed for inactivating of at least 4.0 log units of *B. subtilis* spores (to below detection limits), while HS and HPP caused neither the germination nor inactivated spores (i.e., there was neither a loss of endospore resistance after heat shock nor endospore inactivation as a consequence of the storage methodology). PATP of undiluted honey even at harsh processing conditions (600 MPa/15 min/85 °C) did not impact the spore load. The results for diluted honey open the possibility of its decontamination by spores’ inactivation for medical and pharmaceutical applications.

## 1. Introduction

Honey is a natural viscous product that has received special attention over the years in the setting of novel applications by exploiting its unique composition, including phenolic compounds (e.g., quercetin, kaempferol, catechin, apigenin, chrysin, *p*-coumaric acid, caffeic acid, gallic acid, etc.), enzymes (catalase and superoxide dismutase), sugars, and other trace elements, which are directly linked to health benefits, antioxidant, anti-inflammatory, anticancer, and antiviral activities [1,2,3,4]. Recently, this natural product has been explored in biomedical applications as an antibacterial agent for the development of advanced wound-healing scaffolds (i.e., hydrogels and nanofibrous scaffolds) [5] and honey-based medical formulations such as syrups, eye drops, and pastilles [6]. In addition, honey has a role as a food product preservative, mainly due to its ability to inhibit the growth of spoilage microorganisms and bacteria that cause foodborne diseases [7].

Although microbial contamination of honey is usually low, as its acidity (pH 3.2–4.5) and water activity (a_W_ < 0.65) inhibit microbial growth, several reports have indicated contamination with bacterial spores, mainly of the genera of *Bacillus* and *Clostridium*. *Clostridium botulinum*, a bacterium present in the soil, may be transported by bees in their pollen baskets, nectar, or honeydew to the honeycomb during harvest; thus, in these circumstances, it is impossible to avoid the contamination of honey by the spores. Nevertheless, strict attention to sanitary regulations and hygienic practices during honey harvesting and extraction may significantly reduce the risk of unwanted bacterial contamination [8]. However, despite its rarity, even a single colony-forming unit can produce a neurotoxin that causes botulism symptoms in infants and immunosuppressed individuals [9]. To prevent infant botulism, the World Health Organization (WHO) recommends not using honey as a sweetener in preparations under one year old [10]. Furthermore, as an additional concern, honey may be associated with wound botulism if used as a dressing [11]. Indeed, the spores in honey are in a dormant state, but they can be transmitted when honey is used as an ingredient in another product (i.e., wound dressing), and under certain conditions, (e.g., a_W_ > 0.60) can multiply until the product deteriorates [12], limiting its biomedical application. Therefore, if honey is intended for therapeutic use, it is necessary to guarantee the removal of any pathogens that may present a risk to human health.

Honey pasteurization is employed for preservation, aiming to inactivate vegetative forms of pathogenic and spoilage microorganisms (particularly the yeast), but after pasteurization, honey is not sterile since heat-resistant microbial spores can be present. In honey’s thermal pasteurization, heat treatment with a temperature above 100 °C is not used to achieve commercial sterility, because such an extreme heating temperature can impact honey’s essential composition (phenolic compounds, level of ascorbic acid, enzymes, and sugar) and/or lower its quality (i.e., increase formation of HMF, a quality indicator of honey) [13,14,15]. A recent study showed that even at moderate heating temperatures (62 °C), most of the tested honeys exhibited a loss of antibacterial properties, and at 100 °C, the total loss of bactericidal properties was observed in all tested honeys. Similarly, prolonged thermal processing (2 h), even at non-excessive temperatures (i.e., 42 °C), drastically reduced antibacterial properties [16]. Commonly, commercial medical-grade honey and honey-based wound dressings are sterilized by gamma irradiation to kill *Clostridiu**m* spores [5,17]. However, recent research showed that consumers have a negative perception of irradiated foods [8], and even though the safety results are promising and regulated, many consumers show reluctance towards irradiated foods/products [18]. 

High-pressure processing (HPP) is an emerging, nonthermal (5–20 °C) food processing technology that uses elevated hydrostatic pressures (400–600 MPa) for short periods to inactivate spoilage and pathogenic vegetative microorganisms and certain enzymes [19,20]. The most stringent HPP conditions used industrially are usually 600 MPa for 6 min within the temperature range of 5–30 °C [21], which effectively reduces the occurrence of the Maillard and caramelization reactions, and thus the original color, flavor, quality, and nutrients of food products can be retained to a greater extent compared with conventional processing technologies [22]. The sole use of high pressure has little effect on bacterial spores; thus, a specific combination of high pressure (up to 600 MPa) and temperature (90–120 °C), called pressure-assisted thermal processing (PATP), is applied for the elimination of microbial presence, both vegetative and spores, with a product quality superior to conventional thermal preservation techniques [23]. Hyperbaric storage (HS) is a pressure-based methodology that uses hydrostatic pressure as a hurdle to slow down/inhibit microbial proliferation in foods, as occurs during refrigeration. In HS, foods are to be kept under mild pressures (up to 150 MPa) for the whole storage period, and for extended storage periods, it was verified that gradual microbial inactivation occurs [24], this being an additional advantage compared to refrigeration. 

Hence, although many studies are conducted on endospore inactivation under high-pressure conditions using *Bacillus subtilis* as a model organism in different types of food matrices [23], there are no data on honey. Therefore, this paper aims to investigate if pressure-based approaches, such as nonthermal HPP, PATP, and HS, are feasible for the inactivation of bacterial endospores in honey, using one of the most pressure-resistant Bacilli, such as the *B. subtilis* endospores, as the case study. Additionally, the objective of this work is to determine the minimum water activity (a_W_) of honey required to effectively decontaminate honey by these techniques, as it is reported in the literature that the feasibility of hydrostatic pressure towards microbial inactivation decays by reducing the water activity, i.e., food products with an a_W_ below 0.90 are reported to be unsuitable for HPP [25]. Thermal processing (TP) conditions at 75 and 85 °C for 15 min, which are conventionally applied in honey pasteurization in industry, were carried out for comparison purposes. 

## 2. Materials and Methods

### 2.1. Reagents, Culture Media, and Solutions

HiCrome Bacillus agar was purchased from Sigma-Aldrich (Seelze, Germany), and a physiological solution (0.9% NaCl) was obtained from Panreac AppliChem (Darmstadt, Germany). The glacial acetic acid from Merck (Darmstadt, Germany), acetic anhydride from Panreac (EU), sulphuric acid from JMGS (José Manuel Gomes dos Santos, Lisbon, Portugal), ethanol from Fisher Scientific (Loughborough, UK), and glycerin from VWR chemicals (Fontenay-sous-Bois, France) were used for the pollen analysis.

### 2.2. Botanical Origin of Honey

The honey samples’ pollen was analyzed and executed by the methodology described by Erdtman, (1960) [26]. Ten grams of the sample was dissolved in 20 mL of distilled water, the obtained mixture was centrifuged at 2000× *g* rpm for 5 min, and the supernatant was discarded. After, 5 mL of glacial acetic acid was added and centrifuged at 2000× *g* rpm for 5 min. Then, acetolysis of the pollen sediments was carried out at a mixture of 9:1 of acetic anhydride and sulphuric acid, in a water bath at 100 °C for 2 min. The mixture was centrifuged at 2000× *g* rpm for 5 min, and the supernatant was discarded. After carefully washing the sediment with 5 mL of water containing 3 drops of ethanol and centrifuging it, 5 mL of glycerin-water (50%) was added, and the sediment was mounted in gelatin-glycerinate.

### 2.3. Sample Preparation

The raw (non-diluted) honey samples (15 g) and the diluted honey–water samples (HWP) obtained by adding the necessary amount of water to achieve an aw of 0.80, 0.85, 0.90, and 0.96 (Table 1) were aseptically placed in UV-light-sterilized, low-permeability polyamide–polyethylene, bags (PA/PE-90, Plásticos Macar–Indústria de Plásticos Lda, Palmeiras, Portugal), using a laminar flow cabinet (BioSafety Cabinet Telstar Bio II Advance, Terrassa, Spain) to avoid contaminations, and then thermo-sealed. The a_W_ was measured at 25 °C using a hygrometer (Novasina aw-Sprint, Switzerland), and the results were expressed with two decimals. °Brix was measured using an automatic refractometer (Atago pocket PAL-BX/RI, Japan).

### 2.4. High-Pressure Processing at Room Temperature

Briefly, the whole honey sample and diluted honey samples such as HWP 0.80, HWP 0.85, HWP 0.90, and HWP 0.96 were subjected to high-pressure processing (HPP) at 600 MPa for 15 min using pilot-scale HPP equipment (Hiperbaric 55, Hiperbaric, Burgos, Spain). The input water temperature was 20–22 °C, the compression rate was approx. 250 MPa/min (2500 bar/min), and the decompression time was <5 s. All samples were analyzed immediately after processing.

### 2.5. High Pressure Combined with Thermal Processing

The PATP was performed at 600 MPa for 15 min at 70, 75, and 85 °C in pilot-scale HPP equipment (Hiperbaric 55, Hiperbaric, Burgos, Spain). The temperature selected was the most used for the conventional thermal pasteurization of honey. The samples before processing were preheated by immersing them in a temperature-controlled bath, in which the water acted as a heating medium. The temperature of the water was set at 70, 75, and 85 °C before the PATP to achieve the target temperature in the sample.

Afterward, the pre-heated samples were placed in an insulated basket (made in polypropylene) filled with water (the working fluid) at temperatures of 70, 75, and 85 °C, and then quickly placed in the high-pressure equipment. During the pressurization, adiabatic heating occurs, and Figure 1 illustrates an example of the pressure and temperature histories obtained during the HPP process cycle because of adiabatic heating. During the pressurization phase, the temperature increases adiabatically, and the target temperature is reached by the sample at the end of the pressurization period. The average temperature during the constant pressure phase of the PATP cycle is often considered the processing temperature [27]. Patazca et al. [28] studied the effect of the adiabatic heating during PATP on honey and reported that at an initial temperature of 70 °C, the adiabatic temperature increased by 3.7 °C/100 MPa for the pressure of 581.1 MPa. Thus, it is estimated that the average processing temperature of samples in the present experiment (600 MPa/70, 75, and 85 °C) reached approx. 92.2, 97.2, and 107.2 °C, due to the adiabatic heating. After processing, the insulated container was opened and the temperature measured, displaying 65.3, 71.1, and 82 °C, and the samples were immediately placed in ice-cooled water (4 °C) to cool down. All the honey samples were analyzed immediately after the processing. The effect of PATP at 600 MPa/85 °C, 15 min on the survival of *B. subtilis* endospores was analyzed after the storage for 24 h at controlled temperature of 20 °C.

### 2.6. Hyperbaric Storage

Samples, non-diluted and diluted honey samples (HWP 0.80), were placed in a multivessel pressure system (FPG13900, Stansted Fluid Power, Stansted, UK) at 50, 75, 100, 125, and 150 MPa for up to 60 days of storage at naturally variable/uncontrolled room temperature (RT, 18–23 °C) to infer if long-term exposure to hydrostatic pressure would impact the endospore load inoculated in honey samples. Simultaneously, the control samples were kept at atmospheric pressure (AP) and RT (AP/RT).

### 2.7. Thermal Processing

Thermal inactivation experiments with *B. subtilis* endospores in honey and HWP were carried out at ambient pressure (0.1 MPa) and temperatures of 75 °C and 85 °C for 15 min using a thermostatic water bath (FA 90, FALC Instruments, Treviglio, Italy). For the thermal processing (TP), the samples were fully immersed in the water bath. The come-up time of the center of the packed honey was less than 3 min. After the treatment, the samples were immediately placed in ice-cooled water (4 °C) to cool down. All the honey samples were analyzed immediately after the TP.

### 2.8. Microbiology

#### 2.8.1. Endospore Inoculations

The honey sample was opened under sterile conditions using a laminar flow cabinet (Bio Safety Cabinet Telstar Bio II Advance, Terrassa, Spain) to avoid contaminations. Then, 300 µL of *B. subtilis* endospore suspension was inoculated at a concentration of about 10^4^–10^5^ cells/mL and thermo-sealed. The endospores used in this study were not heat-activated to avoid changes in their pressure resistance, since heat-activated endospores are reported to be more pressure-sensitive [29].

#### 2.8.2. Endospore Preparation

The endospore preparation was performed as described by [30], with minor modifications. *B. subtilis* ATCC 6633 (DSM 347), purchased from Deutsche Sammlung von Mikroorganismen und Zellkulturen (DSMZ, Braunschweig, Germany), was grown in BHI-agar at 30 °C for 24 h. Afterward, a single colony was isolated to obtain an overnight liquid culture, which was spread-plated onto BHI-agar plates and incubated at 30 °C for 24 h. The sporulation was routinely verified by phase-contrast microscopy, taking 15 days to achieve more than 95% of bright-phase endospores. Then, the endospores were harvested by flooding the agar plates with cold (4 °C) sterile distilled water, which was scratched with a bend glass rod. The endospores were then washed with cold sterile distilled water by 3-fold centrifugation (10 min at 5000× *g* at 4 °C). The washed endospores were stored in distilled water and kept in the dark at 4 °C until use.

#### 2.8.3. Determination of Endospore Germination and Inactivation

To assess both germinated (vegetative cells and spores that lost thermal resistance) and non-germinated spores (dormant cells) after each processing condition, an aliquot of honey was heated at 80 °C for 20 min to inactivate vegetative cells [31,32], allowing for the quantification of not only both germinated and non-germinated spores (unheated samples that will be termed the total microbial load, TML), but also non-germinated spores (heated samples, that will be termed as the total endospore load, TEL). Then, decimal dilutions were performed (1.0 g of each sample for 9.0 mL of physiological solution) that were plated in HiCrome Bacillus agar and incubated at 30 °C for up to 72 h. The results were expressed as the decimal logarithm variation, log (N/N0), obtained by the difference between the microbial load after each processing condition (N) and the initial microbial load (N0) before the processing procedures. As the raw honey was not sterilized before the endospore inoculations, it presented endogenous *B. subtilis* loads of approx. 102 cells/g of honey. As such, a differential culture media was used, and only the *B. subtilis* colonies were considered. Moreover, the raw honey was also heat-treated under the aforementioned conditions to infer the presence of endogenous endospores (TEL), which were less than 2.3 log CFU/g of honey of *B. subtilis* endospores.

## 3. Results

### 3.1. Botanical Origin

The melissopalinological analysis showed the presence of *Castanea sativa, Erica australias, Trifolium repens, Quercus suber, Acasia dealbata, Prunus lusitanica, and Cistus ladanifer* pollen grains (Table 2). Honey should contain at least 45% of the corresponding pollen (dominant pollen) to be considered monofloral. Moreover, chestnut honey must be characterized by at least 90% of *Castanea* pollens [33]. In the studied honey, the percentual frequency of the *Castanea* pollen was 53.3%, which was dominantly pollen, but below the minimum percentage of pollen required for the characterization as a chestnut monofloral honey. Thus, the honey used in this study was classified as multifloral honey. The botanical origin influences the chemical composition of honey (i.e., phenolic compounds); hence, this classification impacts its antioxidant, antibacterial, and wound-healing properties [5].

### 3.2. Bacillus Subtilis Endospores’ Response in Honey towards Different HPP Techniques

#### 3.2.1. Pressure-Assisted Thermal Processing

Raw honey had an initial total microbial load (TML) of 2.36 log CFU/mL, which was practically the same as the total endospore load (TEL) after heat shock at 80 °C for 20 min. As such, almost all forms of *B. subtilis* were present in the form of endospores. After inoculation, the *B. subtilis* endospore load increased to 5.65 log CFU/mL, which remained unchanged immediately after PATP (600 MPa, 85 °C, 15 min) and conventional TP (85 °C, 15 min), as observed in Figure 2A and even after 24 h of processing (Figure 2B). As this strategy was not effective to reduce TEL in honey, another pressure-based approach was tested, namely, hyperbaric storage (HS) at room temperature.

#### 3.2.2. Hyperbaric Storage

The effectiveness of HS on spore germination and inactivation was evaluated in the multifloral honey. The initial TEL in raw honey was below quantification limits (2.30 log CFU/g), being 1.66 log CFU/g. After inoculation of about 5 log of *B. subtilis* endospores, the initial TEL in the raw honey reached 5.04 ± 0.01 log CFU/g, as seen in Figure 3A. After 21 days at 75, 100, and 125 MPa at room temperature, no significant changes were found in spore loads, even when the experiments were extended up to 60 days at 125 MPa. In addition, the endospores did not lose thermal resistance during all HS studies, verified by submitting the samples to a heat treatment of 80 °C for 20 min to inactivate vegetative microorganisms and germinated forms of spores (Figure 3B). As the spore loads were similar between the unheated and heated samples, HS was unable to trigger the germination process, contrary to what was verified in other studies at pressures up to 200–300 MPa [22].

### 3.3. B. subtilis Endospores’ Response in Diluted Honey with Adjusted Water Activity after Different HPP Techniques

#### 3.3.1. HPP and PATP of Honey–Water Preparations with Increased a_W_

Considering the limitations observed in the above-described experiments to inactivate *B. subtilis* endospores, and the potential of honey for biomedical applications, the manipulation of the a_W_ of honey and the impact on the inactivation of *B. subtilis* endospores were assessed. To do so, a first screening on the minimal necessary a_W_ (a_W_ values used are displayed in Table 1) to observe endospore inactivation was made, with samples being processed by PATP (at 600 MPa at 75 °C for 15 min), HPP (600 MPa at RT for 15 min), and TP (75 °C for 15 min).

The initial TEL after inoculation was approximately 5 log CFU/g. Neither conventional TP at 75 °C nor nonthermal HPP at 600 MPa, both performed for 15 min, affected the endospore loads in diluted honey with adjusted a_W_ (up to 0.90), as seen in Figure 4. However, the minimal a_W_ necessary to obtain a considerable endospore load reduction was achieved after PATP in HWP with an a_W_ of 0.80, being verified as a reduction of about 1.74 log units. Moreover, the inactivation effect was more pronounced by increasing the value of a_W,_ reaching the detection limit (1.30 log CFU/g) in HWP with a_W_ 0.85 and 0.90 (at least 3.70 log CFU/g of inactivation).

#### 3.3.2. Hyperbaric Storage of Diluted Honey with Adjusted a_W_

Based upon the results obtained for the PATP experiment, a new test involving long-term exposure to hydrostatic pressures by HS, namely, 150 MPa for 21 days, was conducted on an HWP with an a_W_ of 0.80 (HWP 0.80) to imply the possibility of inactivating *B. subtilis* endospores at room temperature, thereby avoiding the use of high temperatures and preventing the degradation of thermolabile compounds. As seen in Figure 5, the HS methodology was not successful in the reduction of *B. subtilis* endospore loads along the storage period, as practically no variation in the microbial and spore loads was observed.

Notwithstanding, HS at uncontrolled room temperatures was quite effective to prevent microbial proliferation, even at an aw as high as 0.80. Conversely, samples stored at atmospheric pressure and room temperature registered heavy microbial development, monitored visually through gas formation (data not shown). This suggests that HS can be a suitable methodology to inhibit microbiological proliferation in diluted honey (aw 0.80), with the advantage of these being at room temperature, with a consequently much lower energetic savings and carbon footprint compared to refrigeration [34].

To minimize the potential formation of HMF that occurs by processing honey at high temperatures, PATP at 600 MPa and 70 °C for 15 min was also tested, as this way would test if using this temperature for HWP would not surpass 100 °C after pressurization due to adiabatic heating could be used to inactivate *B. subtilis* endospores. The samples’ temperature after reaching 600 MPa (initially at 75 °C at atmospheric pressure) is estimated to be between 93 and 105 °C, considering a temperature increase of 3–5 °C due to the adiabatic heating because of the pressure increase (see the Materials and Methods section for details on the pressure adiabatic heating on honey) [28].

As seen in Figure 6, a combination of hydrostatic pressure and temperature resulted in a *B. subtilis* endospore reduction of approximately 2 log units in diluted honey, while for whole honey, no considerable differences were observed between unprocessed (initial) and processed honey. HPP at 600 MPa at room temperature (about 17 °C) increased the endospore load, not due to an increase in the total number of spores inoculated, but instead to the activation process triggered by HPP. Indeed, spore populations are heterogeneous and do not respond the same way to nutrients, considered as those spores that are only able to develop after an intense physical treatment (either heat shock or HPP) to be superdormant [35]. Considering the above-reported results for PATP, this methodology does not ensure the microbial safety of HWP, so temperatures above 70 °C must be used to ensure a safe application of PATP on HWP for both biomedical and pharmaceutical applications.

## 4. Discussion

### 4.1. Survival of B. subtilis Endospores Inoculated in Honey after HPP, PATP, and HS

The effects of high pressure on bacterial spores depend on the applied processing conditions, i.e., the pressure level triggers different germination pathways of endospores, even when combined with thermal processing. For instance, low pressures (up to 150–200 MPa) are more likely to trigger nutrient-like physiological germination (similar to the one that occurs during the nutrient binding to the endospore germination receptors, thus leading to the endospore germination and outgrowth), while pressures above 200–300 MPa are more likely to trigger a non-physiological germination pathway, which results in the direct opening of the dipicolinic acid channels and its release from the endospore core, thus resulting in endospore outgrowth [35,36].

To date, only a few studies report the effect of high pressure on microbial decontamination of honey. Fauzi et al. [37]. processed Manuka honey at 600 MPa for 20 min at 32.60 °C and observed a than a 1 log reduction in *Saccharomyces cerevisiae* [37]. Leyva-Daniel et al. [38] observed a decrease in total aerobic mesophiles and yeast and molds (about 2.4 log units) to below detection limits after HPP at 600 MPa for 15 min at 28–29 °C in Mexican honey. However, the same inactivation levels were not seen for multifloral honey in the present study. The values for the TML (vegetative and dormant cells) in the honey (aw 0.57, 84.60 °Brix) remained unaltered after the HPP at 600 MPa, for 15 min at ambient temperature. Fauzi et al. (2017) [37] stated that increasing honey’s sugar concentrations from 40 to 80 °Brix increases the number of surviving cells. These authors concluded that low microbial inactivation caused by HPP in honey is attributed to the increased sugar content of honey (°Brix, 1 °Brix represents 1 g of sucrose in 100 g of solution), the baroprotective effect of low aw, and the low compressibility of honey [37]. Nevertheless, the resistance of spores to HPP is widely reported in the literature, as this methodology, being a pasteurization methodology, has very little to no effect on bacterial and (some) fungi spores [39,40].

The PATP is an alternative to thermal processing in reaching the inactivation levels required for commercial processing [41], as several studies show the possibility of combining high pressures and temperatures to eliminate bacterial spores, particularly when performed above 100 °C, and thus obtain shelf-stable food products, in a process already approved by the FDA [42,43]. The present study investigated a different *B. subtilis* endospore inactivation approach based on PATP at the pressure of 600 MPa for 15 min and 85 °C. By analyzing Figure 2, the reader may encounter that no overall changes were observed after the PATP (600 MPa, 85 °C, 15 min), nor after the TP (85 °C, 15 min), suggesting a protective effect caused by the low aw of honey, which confers a protective effect to microorganisms, making them more pressure-resistant [44,45]. Data regarding the impact of HPP and PATP in honey, especially in spores, are very scarce, and the results revealed no encouraging effects for spores’ inactivation, i.e., considering the low a_W_ of honey, the protective effect conferred by the carbohydrates on microorganisms, in general, will make a great challenge for the use of pressure-based methodologies for microorganisms’ inactivation [46]. For this, temperatures at and above 121.1 °C for several minutes to destroy bacterial spores in honey are needed, to avoid safety issues [47]; however, such conditions deteriorate the antibacterial properties of honey [16].

When it comes to hyperbaric storage, Pinto et al. [48] reported a gradual decrease of *B. subtilis* endospores in carrot juice (approximately 4 log units of reduction after 60 days at 100 MPa), while in the present study, HS did not affect the endospore loads of the same microorganism, which, as mentioned above, can be related to the low aw of honey that exerts a protective effect against pressure inactivation. Previous works with HS revealed that this preservation methodology can inhibit microbial development (in a range of 60–75 MPa) or even inactivate (at and above 75 MPa) microorganisms in foods [24], which was not observed for non-inoculated samples of honey in the present study.

The low pressures in HS (up to 150 MPa and 60 days) and the high-pressure level (600 MPa) with or without temperature (85 °C) did not trigger germination pathways of *B. subtilis* endospores, nor inactivation in multifloral honey.

### 4.2. Inactivation of Vegetative Cells and Endospores of B. subtilis in Diluted Honey with Increased a_W_ Using HPP, PATP, and HS

Usually, the feasibility of HPP is limited to food products whose a_W_ is equal to or greater than 0.90 [44], which is evidenced by the ineffectiveness of the HPP in honey samples with an a_W_ of 0.60 and 0.90. In the present study, little to no impact on endospore inactivation was observed, regardless of the water activity level (up to 0.90) of the diluted honey after the HPP (600 MPa for 15 min at RT), as well as TP (75 °C for 15 min), as expected. Indeed, nonthermal HPP and conventional thermal pasteurization are unable to inactivate bacterial (and some fungal) spores, due to their dehydrated state and exquisite set of heat-shock proteins that protect them against HPP and thermal pasteurization, respectively [31,39,49].

Differently, PATP at 600 MPa for 15 min at 75 °C reduced TEL in approximately 2 log units at a minimum a_W_ of 0.80, indicating a need for using higher temperatures to achieve greater endospore loads’ inactivation to ensure food safety and the safe application of honey-based preparations for (bio)medical applications. The inactivation effect was greater at higher water activities (a_W_ 0.85 and 0.90), reaching the detection limit of 1.30 log CFU/g, as seen in Figure 4. Indeed, the effectiveness of PATP for endospore inactivation is quite dependent on the water activity and temperature of the food matrix, as higher water activities will increase pressure transmission and decrease the protective effect that the solutes provide to the microorganisms and their spores [50].

A combination of hydrostatic pressure and temperature could be a feasible alternative to conventional sterilization processes to destroy bacterial spores, with the major advantage of using lower temperatures than those employed in fully thermal sterilization processes [51]. Yet, this may not be the case for some *Clostridia* spp., as these may require higher temperatures for a successful inactivation process, such as 93–95 °C, to achieve temperatures between 110–125 °C, ensuring an efficient inactivation process and, thus, assuring food safety [52].

Even though this approach could be undeniably pertinent to enhancing the bioactivity of honey, a successful commercial application of pressure combined with high temperatures has yet to be implemented in the food industry, due to technical reasons related to proper HPP equipment able to ensure homogeneous pressure vessel heating and temperature uniformity accordingly [53], but PATP for food sterilization is already approved by the FDA [54].

No appreciable changes were observed in the HS of diluted honey with an a_W of_ 0.8 throughout storage, indicating that long-term exposure to hydrostatic pressure (in the range of HS conditions, at 150 MPa for 21 days) was ineffective against bacterial spores (Figure 6), as previously reported for other cases [48]. However, Pinto et al. [48] evaluated the effectiveness of HS to control the development of *B. subtilis* endospores and observed that a low pressure of 50 MPa would result in a 4-log reduction along 60 days, at room temperature, unlike in the present work, wherein neither germination nor inactivation was observed. Indeed, this can be attributed to the low a_W_ of the honey that confers a protective effect against hydrostatic pressure and would also hinder the germination process (usually does not occur at a_W_ values below 0.90) [55].

## 5. Conclusions

The present study evaluated the impact of pressure-based strategies and water activity on the inactivation of honey bacterial spores, using *B. subtilis* as a case study. As expected, nonthermal HPP had little to no effect on honey’s microbial loads, yet this methodology could still be used to enhance honey’s bioactivity, as demonstrated in other studies.

Regarding the presented methodologies, pressure-assisted thermal processing (PATP) seems to be the most promising technique for the destruction of bacterial spores, especially at moderate/high water activity (0.85 onwards), wherein it was possible to inactivate bacterial spores below detection limits (1.30 log CFU/g). Hyperbaric storage at uncontrolled room temperature is an adequate preservation methodology, as it hindered the development of bacterial spores, with lower energetic costs and a lower carbon footprint. Despite this, further and deeper studies are needed to validate the use of these pressure-based methodologies in the setting of biotechnological honey-based preparations, such as hydrogels and films.

## Figures and Tables

**Figure 1 molecules-27-05918-f001:**
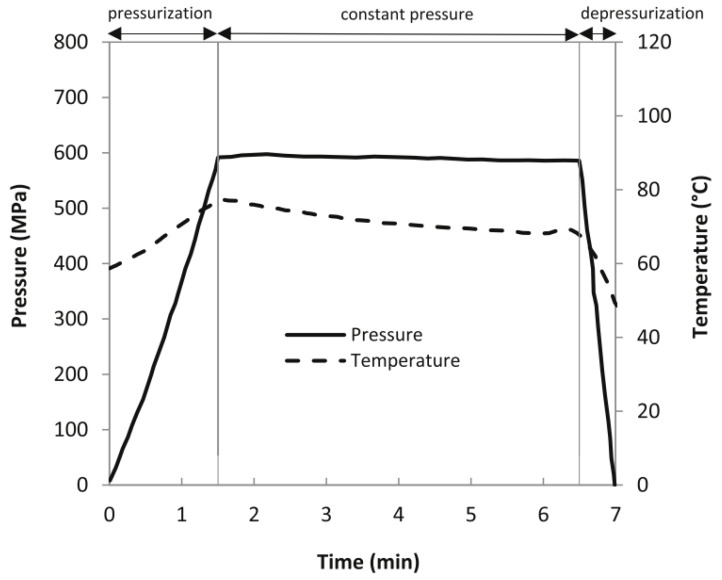
An example of the pressure and temperature (P–T) history of a PATP process (at 600 MPa, 70 °C, for 5 min). Adapted with permission [27].

**Figure 2 molecules-27-05918-f002:**
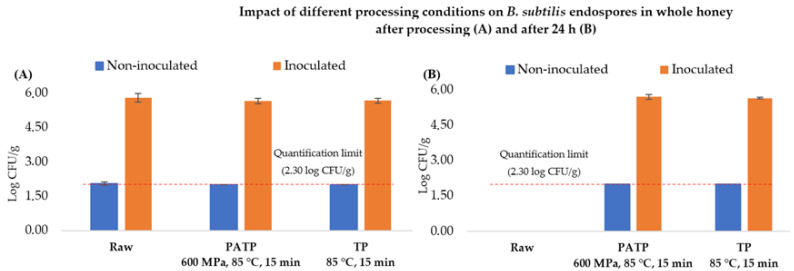
Effects of pressure-assisted thermal pasteurization (PATP) and conventional thermal pasteurization (TP) on the survival of total *B. subtilis* microbial load (TML) in non-inoculated honey (blue bars) and the total *B. subtills* endospore load (TEL) in inoculated honey (orange bars): (**A**) immediately after processing (day 0), and (**B**) after 24 h of storage of the processed honey. Dashed bars mean that the quantification limit of 2.30 log CFU/g was reached.

**Figure 3 molecules-27-05918-f003:**
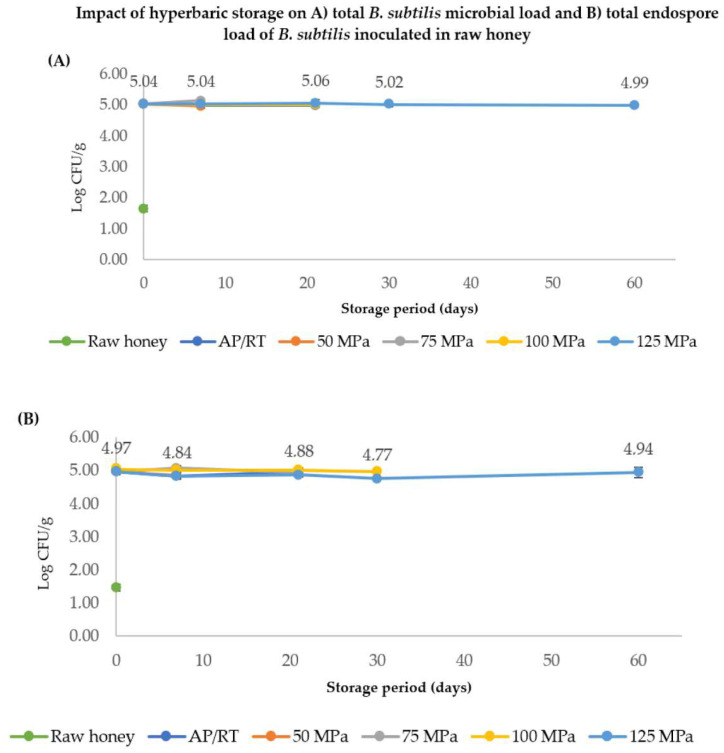
Effects of hyperbaric storage at 50, 75, 100, and 125 MPa at room temperature (20–23 °C): (**A**) on the total *B. subtilis* microbial load (vegetative and endospores cells) in honey, and (**B**) on the *B. subtilis* endospores (after heat-shock at 80 °C for 20 min). AP/RT regards control samples stored at the same room temperature (20 °C) but at atmospheric pressure, while raw honey regards the indigenous *B. subtilis* load and endospores naturally present in honey.

**Figure 4 molecules-27-05918-f004:**
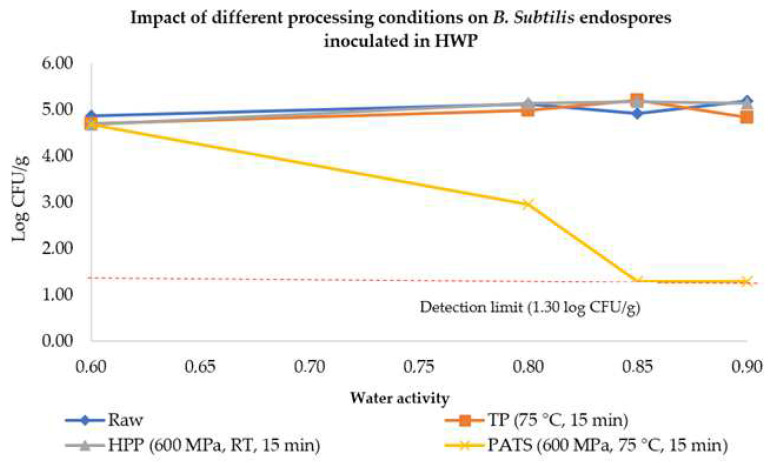
Impact of HPP (600 MPa for 15 min at room temperature), PATP (600 MPa, 75 °C, 15 min), and conventional thermal pasteurization (TP) (75 °C for 15 min) on the survival of *B. subtilis* endospores in whole (a_W_ = 0.60) and HWP (honey–water preparations) with an adjusted water activity (a_W_ up to 0.90).

**Figure 5 molecules-27-05918-f005:**
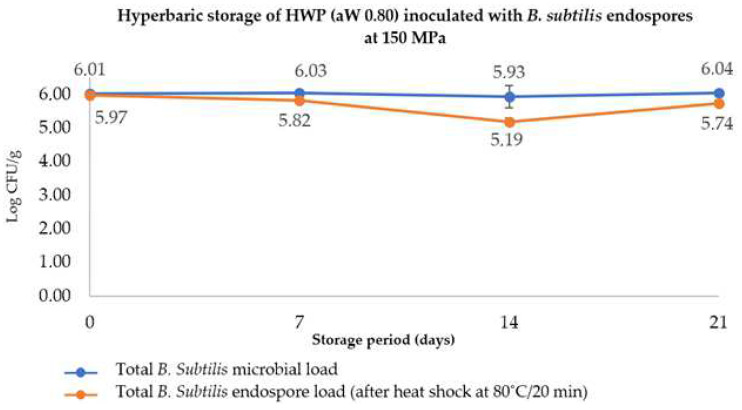
Effect of hyperbaric storage (HS) of inoculated HWP (honey–water preparation) with an adjusted water activity (a_W_ = 0.80) at 150 MPa at uncontrolled room temperature (20–23 °C) up to 21 days on the total *B. subtilis* microbial load (blue lines) and endospore load (orange lines, after a heat-shock of 80 °C for 20 min).

**Figure 6 molecules-27-05918-f006:**
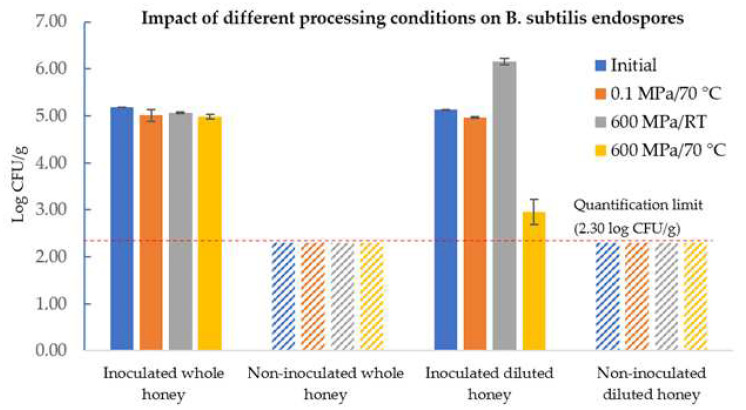
Effect of different processing conditions (thermal pasteurization, 70 °C/15 min), nonthermal HPP (600 MPa/15 min, RT), and PATP (600 MPa/15 min/70 °C) on *B. subtilis* endospores inoculated in HWP with an a_W_ of 0.96. Dashed bars mean that the loads were below the quantification limit of 2.30 log CFU/g.

**Table 1 molecules-27-05918-t001:** Honey and honey–water preparations (HWP) with an adjusted water activity (a_W_).

Sample	°Brix	a_W_
Honey (raw, non-diluted)	84.6	0.57
Diluted honey		
HWP 0.80	63.3	0.80
HWP 0.85	52.9	0.85
HWP 0.90	43.4	0.90
HWP 0.96	33.9	0.96

**Table 2 molecules-27-05918-t002:** Botanical origin of honey: dominant pollen (D) >45% of the pollen spectrum; accompanying pollen (A) representing 15–45%; important pollen (I) 3–15%; minor pollen (R) 1–3%.

Quantitative Pollen Spectrum (%)				
D	A	I	R	Type of Honey
*Castanea sativa*	*Erica australias*	*Trifolium repens*	*Cistus ladanifer*	Multifloral
(53.3%)	(19.1%)	(12.8%)	(2.9%)	
		*Quercus suber*		
		(4.3%)		
		*Acasia dealbata*		
		(3.8%)		
		*Prunus lusitanica*		
		(3.8%)		

## Data Availability

Not applicable.

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
