# Peer review of "High-Pressure-Based Strategies for the Inactivation of *Bacillus subtilis* Endospores in Honey"

_molecules, 2022, doi:10.3390/molecules27185918_

Round 1
Reviewer 1 Report
High Pressure-Based Strategies for the Inactivation of Bacillus subtilis Endospores in Honey
Comments:
Line no 10: Mention here the main bioactive compounds.
Line no 34: What main component made its unique composition?
Line no 43: Does natural honey contains these spores or found during processing?
Line no 58: What will be the effect of thermal treatment on nutritional content of honey?
Line no 64: Irradiation process is completely safe with no adverse impacts?
Line no 69: Add the references for this statement regarding HPP, please consult with these papers for updated data
· High-pressure processing for sustainable food supply
· High-Pressure Treatments for Better Quality Clean-Label Juices and Beverages: Overview and Advances
Line no 97: Mono floral honey will have more nutritional value or multi floral?
Line no 103. it is comma or dot between all the digit mentioned in the Table 1? Please confirm.
Line no 118. Red font should be replaced with black font. First letter should be capital for Inoculated and non-inoculated.
Line no 142: Graphical representation is appreciable.
Line no 168. Red font should be replaced with black font
Line no 192: What is the relation between temperature and HS?
Line no 196: How more water activity will hinders microbial growth in HS?
Line no 218. Red font should be replaced with black font
Line no 325: Materials and methods section should be before results and discussion.
Line no 453: Conclusion should be rewritten in more comprehensive manner.
· Overall this manuscript is very well written, only few minor points are needed to be cleared.
Author Response
The authors acknowledge the referees’ time in considering our manuscript. The manuscript was thoroughly reviewed, and more information was added as requested. Please check the manuscript with the highlighted changes made in the red-color text. Please, find our point-by-point response to the comments and suggestions in the attachment.

Reviewer 2 Report
The authors of this manuscript molecules-1876941 entitled ‘High Pressure-Based Strategies for the Inactivation of Bacillus subtilis Endospores in Honey’ demonstrated the efficacy of high-pressure processing in the inactivation of Bacillus subtilis in honey. This is novel research, and the results will be interesting for the readers of the Molecules. However, there are many factors that preclude the acceptance of this manuscript in its current form. Overall, I think the grammar used in this manuscript should be improved.
Abstract.
The introductory part of the abstract is too lengthy and the major information the authors were trying to convey was lost. I suggest rewriting this section in a concise manner.
Introduction
The authors should try and connect the paragraphs in this section in addition to improving the grammar
Materials and methods:
Can the authors comment on why it is essential to dilute the honey before it is treatment? I suggest that this information should be included in the introduction also.
References
The authors didn’t follow the guidelines for the authors in the reference listing, which boils down to the ability to pay attention to details. Please correct and do the needful.
Author Response
The authors acknowledge the referees’ time in considering our manuscript. The manuscript was thoroughly reviewed, the English language was improved, and more information was added as requested. Please check the revised manuscript with the highlighted changes made in the red-color text. Please, find our point-by-point response in the attachment.

Reviewer 3 Report
Minor concerns:
Comments to the Author:
The manuscript's title is appropriate.
Abstract: The Background of the abstract is well written. The main procedure and findings of the study are well expressed.
Introduction: A brief survey of existing literature, purpose, importance, and innovation of the research is well mentioned. The conclusion is short and to the point.
English is not my native language. Having that into consideration, I found the writing difficult to read and lacking fluidity between the subjects approached.
Please find major comments in the attached file.

Author Response
The authors acknowledge the referees’ time in considering our manuscript. The manuscript was thoroughly reviewed, and more information was added as requested. Please check the manuscript with the highlighted changes made in the red-color text. Please, find the point-by-point response to your comments in the attached file.

Round 2
Reviewer 2 Report
Accept